# Preparation of Hollow Core–Shell Fe_3_O_4_/Nitrogen-Doped Carbon Nanocomposites for Lithium-Ion Batteries

**DOI:** 10.3390/molecules27020396

**Published:** 2022-01-08

**Authors:** Jie Wang, Qin Hu, Wenhui Hu, Wei Zhu, Ying Wei, Kunming Pan, Mingbo Zheng, Huan Pang

**Affiliations:** 1School of Chemistry and Chemical Engineering, Yangzhou University, Yangzhou 225002, China; wj1159785@163.com (J.W.); huqinchemistry@foxmail.com (Q.H.); huwenhui0919@163.com (W.H.); zhuwei_77@163.com (W.Z.); 13206588726@163.com (Y.W.); 2Hengshanqiao Senior Middle School, Wujin District, Changzhou 213119, China; 3National Joint Engineering Research Center for Abrasion Control and Molding of Metal Materials & Henan Key Laboratory of High-Temperature Structural and Functional Materials, Henan University of Science and Technology, Luoyang 471003, China; 4College of Materials Science and Technology, Nanjing University of Aeronautics and Astronautics, Nanjing 210016, China

**Keywords:** hollow core–shell structure, iron oxide, lithium-ion batteries

## Abstract

Iron oxides are potential electrode materials for lithium-ion batteries because of their high theoretical capacities, low cost, rich resources, and their non-polluting properties. However, iron oxides demonstrate large volume expansion during the lithium intercalation process, resulting in the electrode material being crushed, which always results in poor cycle performance. In this paper, to solve the above problem, iron oxide/carbon nanocomposites with a hollow core–shell structure were designed. Firstly, an Fe_2_O_3_@polydopamine nanocomposite was prepared using an Fe_2_O_3_ nanocube and dopamine hydrochloride as precursors. Secondly, an Fe_3_O_4_@N-doped C composite was obtained by means of further carbonization treatment. Finally, Fe_3_O_4_@void@N-Doped C-*x* composites with core–shell structures with different void sizes were obtained by means of Fe_3_O_4_ etching. The effect of the etching time on the void size was studied. The electrochemical properties of the composites when used as lithium-ion battery materials were studied in more detail. The results showed that the sample that was obtained via etching for 5 h using 2 mol L^−1^ HCl solution at 30 °C demonstrated better electrochemical performance. The discharge capacity of the Fe_3_O_4_@void@N-Doped C-5 was able to reach up to 1222 mA g h^−1^ under 200 mA g^−1^ after 100 cycles.

## 1. Introduction

With the rapid development of the global economy, the shortage of fossil fuels and the worsening of environmental pollution have become a great threat to mankind. Therefore, people must develop green and environmentally friendly clean energy to replace traditional fossil fuels [1,2,3,4]. Solar, wind, and tidal energy are renewable and result in lower levels of pollution, making electricity generated from them a good alternative [5,6,7,8]. However, these sources are often intermittently limited. Electrochemical energy storage provides a feasible way to store electric energy [9,10,11,12]. On the one hand, it can solve the intermittence problem of the above energy sources. On the other hand, this allows the mobile storage of energy. Among various electrochemical energy storage devices, lithium-ion batteries have attracted more and more attention because of their high energy density, long life cycle, and environmental friendliness [13,14,15,16,17].

As an important branch of anode materials for lithium-ion batteries, transition metal oxides (M*_x_*O*_y_*, M = Fe, Co, Ni, Cu, Mn, etc.) have attracted more and more attention. The conversion reaction between transition metal oxides and lithium ion means that they have a high lithium storage capacity [18,19,20,21,22]: M*_x_*O*_y_* + 2*y*Li^+^ + 2*y*e^−^ = *y*Li_2_O + *x*M. In general, transition metal oxides have a significantly higher theoretical specific capacity than graphite when used as anode materials for lithium-ion batteries [23,24,25]. However, it is inevitable that transition metal oxides will experience large volume deformation during the cycling process, which leads to the pulverization of the electrode materials and also results in serious capacity fading [26,27,28,29]. Researchers have started to pay a great deal of attention to one of the transition metal oxides, iron oxide (FeO*_x_*). Compared to other metal oxides, such as CoO*_x_* and NiO, FeO*_x_* has the advantages of being a cheap and abundant resource that is also environmentally friendly [30,31,32]. However, the volume expansion effect and low electrical conductivity limit the application of FeO*_x_* in production [30,33,34,35].

Forming composite materials with a carbon coating is an effective way to improve the performance of iron oxides [36,37]. On the one hand, carbon coating can improve the electrical conductivity of electrode materials. On the other hand, it can alleviate the volume expansion effect of iron oxides during the cycling process [36,37]. The conductivity of carbon materials obtained via the carbonization of common carbon sources is usually not good. By doping carbon with nitrogen, the conductivity of carbon materials can be greatly improved in order to better improve the electrochemical performance of iron oxides [38,39,40,41,42,43,44,45,46]. In addition, relevant studies show that the nano-cavity structure can accommodate the volume deformation of the electrode material well during the cycling process, thus improving the cycling performance of the material [47,48]. In this work, Fe_2_O_3_ cubes with uniform size are prepared by means of the hydrothermal method. Then, polydopamine is coated on the surface of Fe_2_O_3_ cubes. After carbonization, Fe_3_O_4_@nitrogen-doped carbon (N-doped C) composites are formed (Figure 1). The composite is further etched with hydrochloric acid to prepare the core-shell Fe_3_O_4_@void@N-doped C nanocomposites (Figure 1). The void size of the nanocomposites can be regulated via controlling the etching time, improving the performance of FeO*_x_* so that it can be used as the electrode material in lithium-ion batteries. A series of Fe_3_O_4_@void@N-doped C-*x* nanocomposites (*x* is 2, 5, and 10, representing the etching time of 2 h, 5 h and 10 h, respectively) were obtained by etching the carbonized product in acid different numbers of times.

## 2. Results and Discussion

Appendix A shows the X-ray diffraction (XRD) pattern of the Fe_2_O_3_ precursor. Compared to the standard cards, the characteristic peaks of the Fe_2_O_3_ cube samples are consistent with the standard peaks of α-Fe_2_O_3_ (JCPDS:33-0664). Appendix A shows the XRD pattern of an Fe_3_O_4_@void@N-doped C-2 composite. Compared to the standard card (JCPDS:19-0629), it can be seen that the main characteristic peaks of Fe_3_O_4_@void@N-doped C-2 are consistent with the standard peaks of Fe_3_O_4_, and the impurity peaks of FeO also appear. This shows that during the carbonization process, most of the Fe_2_O_3_ is converted into Fe_3_O_4_ and that some of it is converted into FeO.

Figure 1a,b and Appendix A are scanning electron microscope (SEM) and transmission electron microscope (TEM) images of an Fe_2_O_3_ precursor. A large area of cubes with an average size of about 600 nm can be clearly observed. Figure 1c and Appendix A are SEM images of an Fe_2_O_3_@PDA composite material. It can be observed that the Fe_2_O_3_ cube is coated with a dopamine nanolayer. Figure 1d is a TEM image of the Fe_2_O_3_@PDA cube. It is more obvious that the Fe_2_O_3_ cube is coated with a dopamine nanolayer, and the average thickness of the coated PDA layer is about 25 nm. Figure 1e,f and Appendix A are SEM and TEM images of the heat-treated Fe_3_O_4_@N-doped C sample. It can also be clearly observed from Figure 1e and Appendix A that voids appear in each cube. This is because the Fe_2_O_3_ cube is converted into Fe_3_O_4_ and FeO under the reductive atmosphere that is generated by dopamine in the process of high temperature carbonization, resulting in volume shrinkage. From the TEM image in Figure 1f, the carbon layer structure and hollow groove structure on the surface of the cube can be observed more clearly.

Figure 2a,b are the SEM and TEM images of Fe_3_O_4_@void@N-doped C-2. It can be seen that the void in the cube is obviously larger than the carbonization product before treatment. After 5 h of etching under the same conditions, it can be observed that the void in the cube becomes larger, that some Fe_3_O_4_ particles present a hollow structure, and that the core–shell structure is more obvious (Figure 2c,d). After 10 h of etching, it can be observed from Figure 2e,f that the void in the cube continued to grow. This shows that the longer the etching time in hydrochloric acid is at a certain concentration, the larger the corresponding void will be. In addition, Figure 2g,h show that the internal Fe_3_O_4_ of the Fe_3_O_4_@N-doped C composite has been completely etched after 2 h of etching with 4 mol L^−1^ hydrochloric acid at 30 °C, forming a hollow carbon cube structure. The above results show that the overall morphology of the material basically does not change before and after heat treatment, and the external carbon layer can still cover the Fe_3_O_4_ particle. After a certain etching period, the core–shell structure is formed. Thermogravimetric analysis (TG) was used to detect the carbon content of the composites (Appendix A). It can be seen from the test results that the mass increases from about 225 °C. This is because Fe_3_O_4_ is gradually oxidized into Fe_2_O_3_ as the temperature increases. The apparent subsequent weightlessness corresponds to the oxidation of carbon to CO_2_. After the calculation, the carbon contents of the Fe_3_O_4_@void@N-doped C-*x* composites (*x* = 2, 5 and 10) were determined to be 6.3%, 11.7%, and 20.0%, respectively.

Figure 3 shows the energy dispersive spectroscopy (EDS) element mapping analysis results for the Fe_3_O_4_@void@N-doped C-5 composite. The results show that the Fe_3_O_4_@void@N-doped C-5 composite contains four elements: Fe, O, N, and C. Figure 3d–i show that the carbon layer completely wraps the inner Fe_3_O_4_, which is also effective in proving the existence of the core-shell structure. There is an obvious gap between the core and the shell, which is consistent with previous SEM and TEM results. In addition, Figure 3e,f show that N is evenly distributed in the carbon shell, which indicates that nitrogen doping is achieved when PDA is used as a carbon source. In addition, the electron diffraction image (Figure 3c) shows that the Fe_3_O_4_ core has a single crystal structure.

Figure 4a shows the Raman spectra of the Fe_3_O_4_@void@N-doped C-*x* composite. Raman peaks appear at the positions at about 1300 and 1580 cm^−1^. The peak around 1300 cm^−1^ is known as the D peak for carbon, which is caused by atomic lattice defects on the surface of the carbon material. The peak at about 1580 cm^−1^ is known as the G peak of carbon, which is formed by the graphitization of carbon material. The appearance of the D peak and the G peak shows that the carbon materials that were formed by dopamine carbonization have internal defects and a certain degree of graphitization. The strength ratio (*I_D_*:*I_G_*) of the two peaks of the three composite materials was about 0.65. Because the three composite materials were formed via the etching of Fe_3_O_4_@N-doped C samples at different times, the strengths of peak D and peak G is not affected. In addition, very weak Raman characteristic peaks of Fe_3_O_4_ also appear at about 200 and 300 cm^−1^.

Figure 4b–f show the X-ray photoelectron spectroscopy (XPS) analysis results of the Fe_3_O_4_@void@N-doped C-5 composite. The presence of Fe, O, C, and N can be observed from Figure 4b, which proves that the sample is composed of Fe, O, C, and N elements. Two characteristic peaks of Fe 2p can be clearly observed at 711.1 eV and 724.5 eV, respectively (Figure 4c). By calculating the peak area in the spectrogram, it can be known that the ratio of Fe^3+^ to Fe^2+^ is about 2:1, which is in agreement with the ratio of ferric iron to ferrous iron in Fe_3_O_4_. For the O 1s spectrogram (Figure 4d), there are three obvious characteristic peaks in the spectrogram at about 530.4, 532.3 and 533.7 eV, respectively, which correspond to the XPS peak of O^2−^. For the C 1s spectrogram (Figure 4e), the three smooth peaks at about 284.9, 286.6, and 289.1 eV correspond to C-C, C-N and C=O bonds, respectively. The characteristic peaks of graphite nitrogen and pyridine nitrogen can be clearly observed in the N 1s spectrogram (Figure 4f) and correspond to 400.8 eV and 398.5 eV, respectively. Through XPS analysis, it can be further proven that nitrogen-doped carbon is formed after the carbonization of PDA.

Figure 5a and Appendix A show the cyclic voltammetry (CV) curves of the Fe_2_O_3_ and Fe_3_O_4_@void@N-doped C-*x* composite materials (*x* = 2, 5 and 10) with a voltage range of 0.01–3.0 V and a voltage scanning speed of 0.5 mV S^−1^. In Appendix A, there is an obvious reduction peak at 0.43 V in the first cycle of the CV curve. The reduction peak here is due to the reduction of Fe_2_O_3_ to Fe and the formation of Li_2_O (Fe_2_O_3_ + 6Li^+^ + 6e^−^→2Fe + 3Li_2_O). Due to the solid electrolyte interface (SEI) membrane formed by some of the materials and some of the electrolytes at the same time, the position and intensity of the reduction peak in the following two cycles demonstrate obvious changes [49,50,51]. In the CV curve, the oxidation peak appears at 1.72 V, where the oxidation process is the oxidation of Fe to Fe^3+^. In the following two cycles, the oxidation peak appears to have been obviously attenuated, indicating that the cycling performance of the Fe_2_O_3_ cube sample is poor. Figure 5a and Appendix A show the CV curves of the composites of the Fe_3_O_4_@void@N-doped C-*x*. The main active material in the three composites is Fe_3_O_4_, and the reduction reaction that takes place during the first cycling is Fe_3_O_4_ + 8Li^+^ + 8e^−^ →3Fe + 4Li_2_O. The difference between the three samples is only the void size of the core–shell structure. Therefore, their CV curves are basically same. The reduction peak of the Fe_3_O_4_@void@N-doped C-2 in the first cycle is at 0.36 V and tends to be stable after shifting to 0.8 V in the subsequent cycle. The corresponding oxidation peak is at 1.82 V and tends to be stable after shifting to 2 V in the subsequent cycle. In addition, the change trend in the redox peak in the Fe_3_O_4_@void@N-doped C-5 and Fe_3_O_4_@void@N-doped C-10 cycles is similar to that of Fe_3_O_4_@void@N-doped C-2.

Appendix A shows the charge–discharge curve of the cubic Fe_2_O_3_. The current density is 200 mA g^−1^, and the tested charge–discharge voltage range is 0.01–3 V. It can be observed that the specific capacities of the first discharge and charge cycle of the cubic Fe_2_O_3_ are 1064.8 mA h g^−1^ and 690.7 mA h g^−1^, indicating that the cubic Fe_2_O_3_ material has a large irreversible capacity for the first cycle. At the 2nd, 10th, and 50th cycles, the discharge specific capacities are 72.6.4, 373.9, and 143.5 mA h g^−1^, respectively. The cycle capacity decreases rapidly, indicating that the cycle performance of the cubic Fe_2_O_3_ is very poor. Figure 5b and Appendix A show the representative charge–discharge curves of the first 50 cycles of the Fe_3_O_4_@void@N-doped C-*x* (*x* = 2, 5, 10) at a 200 mA g^−1^ current density. It can be observed that the first discharge platform of the three materials is about 0.75 V, which corresponds to the position of a reduction peak in the CV curve. The initial specific discharge capacities of the Fe_3_O_4_@void@N-doped C-2, Fe_3_O_4_@void@N-doped C-5, and Fe_3_O_4_@void@N-doped C-10 are 1155.8, 1302.4, and 1185.9 mA h g^−1^, respectively. The specific charging capacities are 827.3, 903.6 and 807.8 mA h g^−1^, respectively, which are lower than the corresponding specific discharge capacities, indicating that the composite also has a specific first irreversible capacity. Relative to the specific initial discharge capacity, the irreversible capacity loss is mainly due to the formation of SEI films. Through the comparison of the four samples, it can be seen that the initial discharge capacity of the composite materials is higher than that of the pure cubic Fe_2_O_3_, which is due to the nitrogen-doped carbon significantly improving the conductivity of the composite materials. After 50 cycles, the charge–discharge capacity of the cubic Fe_2_O_3_ is significantly lower than that of the Fe_3_O_4_@void@N-doped C-*x* composite, and the capacity fading phenomenon is very serious. However, Fe_3_O_4_@void@N-doped C-*x* still maintains a high specific capacity after 50 cycles, indicating that the cycling performance of the inner Fe_3_O_4_ core is significantly improved under the protection of the carbon shell. Moreover, after the same 50 cycles, the specific discharge capacities of Fe_2_O_3_, Fe_3_O_4_@void@N-doped C-2, Fe_3_O_4_@void@N-doped C-5, and Fe_3_O_4_@void@N-doped C-10 are 143.5, 825.8, 1132.9, and 730 mA h g^−1^, respectively. With the increase in void size, the specific capacity of the composites does not increase correspondingly, and the Fe_3_O_4_@void@N-doped C-5 material with a suitable core–shell void size shows the maximum specific capacity. The specific capacity of the Fe_3_O_4_@void@N-doped C-10 is lower than that of Fe_3_O_4_@void@N-doped C-5. This is probably due to the fact that the relative content of active Fe_3_O_4_ in Fe_3_O_4_@void@N-doped C-10 is lower than that of the Fe_3_O_4_@void@N-doped C-5 sample. For Fe_3_O_4_@void@C-2, although its Fe_3_O_4_ content is very high, the high Fe_3_O_4_ content will result in the composite having low overall conductivity. Therefore, its specific capacity is also lower than that of Fe_3_O_4_@void@N-doped C-5.

Figure 5c shows the cycle performance diagram of the cube Fe_2_O_3_ and the Fe_3_O_4_@void@N-doped C-*x*. The specific discharge capacities of the cubic Fe_2_O_3_, Fe_3_O_4_@void@N-doped C-2, Fe_3_O_4_@void@N-doped C-5, and Fe_3_O_4_@void@N-doped C-10 are 143.3, 601.4, 1222, and 802.9 mA h g^−1^ after 100 cycles, respectively, when the current density is 200 mA g^−1^ and when the charge and discharge voltage range is 0.01–3V. Comparing the four materials, the specific capacity of the carbon-coated composites is significantly higher than that of the simple Fe_2_O_3_ material. Among them, Fe_3_O_4_@void@N-doped C-5 has the highest specific capacity, and its capacity continues to increase during the cycle, which is mainly caused by the decomposition of electrolyte. After 60 cycles, the capacity of the Fe_3_O_4_@void@N-doped C-2 gradually decreases. This is probably due to the small size of the void between the core and shell of the composite material, which cannot completely alleviate the volume deformation of the Fe_3_O_4_ core during the charging and discharging process. With the increasing number of cycles, the ability of the shell to control the volume deformation of Fe_3_O_4_ gradually weakens. However, Fe_3_O_4_@void@N-doped C-5 and Fe_3_O_4_@void@N-doped C-10 show good cyclic stability due to their large internal void size.

The rate performance of the cubic Fe_2_O_3_ and Fe_3_O_4_@void@N-doped C-*x* (*x* = 2, 5 and 10) composites were further tested (Figure 5d). At the current densities of 100, 200, 400, 800, 1000, and 1600 mA g^−1^, the discharge specific capacities of Fe_3_O_4_@void@N-doped C-5 were determined to be 1010.2, 955.3, 889.6, 823.6, 833, and 735.5 mA h g^−1^, respectively. Although the specific capacity of the Fe_3_O_4_@void@N-doped C-5 composite decreased as the current density increased, it still showed a high specific capacity at 1600 mA g^−1^, indicating that the composite has a good rate performance. When the current density decreased to 100 mA g^−1^ again, the specific discharge capacity was able to increase to 1165.8 mA h g^−1^.

In addition, cyclic tests were also carried out for the Fe_2_O_3_ and Fe_3_O_4_@void@N-doped C-*x* composites at 800 mA g^−1^ (Appendix A). After 100 cycles, the specific discharge capacities of the cubic Fe_2_O_3_, Fe_3_O_4_@void@N-doped C-2, Fe_3_O_4_@void@N-doped C-5, and Fe_3_O_4_@void@N-doped C-10 materials were 67.8, 188, 602.5 and 506.6 mA h g^−1^, respectively. Compared to the cycle diagram at 200 mA g^−1^, the cycle performance at the high current density of 800 mA g^−1^ is inferior to that at the low current density of 200 mA g^−1^. In general, lithium-ion battery electrode materials often show better cycle performance at high current densities. However, for the Fe_3_O_4_@void@N-doped C-*x* composite materials, an abnormal phenomenon occurs. This is probably because the active material in the composite system is a single Fe_3_O_4_ crystal material with large particles, which is not suitable for charging and discharging with large currents. The electrochemical performance of the Fe_3_O_4_@void@N-doped C-5 was compared to the results from reports that have previously been published in the literature. The specific capacity and cycling performance of the Fe_3_O_4_@void@N-doped C-5 at 200 mA g^−1^ reached the level of other similar Fe_3_O_4_-based anode materials [52,53].

By comparing the related electrochemical test data above, among all of the composites, the Fe_3_O_4_@void@N-doped C-5 composite has the best cycling performance and rate performance. This is directly related to the appropriate acid etching time. The proper pore size between the carbon shell and the Fe_3_O_4_ core in the Fe_3_O_4_@void@N-doped C-5 material (larger than Fe_3_O_4_@void@N-doped C-2) can effectively alleviate the volume expansion of Fe_3_O_4_ during the cycle and can also facilitate electrolyte diffusion and the transmission of lithium ions. In addition, the relatively high content of the Fe_3_O_4_ active substance in the material (more than in Fe_3_O_4_@void@N-doped C-10) ensures its large specific capacity. After 100 cycles, the electrode based on Fe_3_O_4_@void@N-doped C-5 was characterized via SEM and TEM (Appendix A). The results show that the structure is essentially retained after 100 cycles. This indicates the good structural stability of the material during the charge/discharge process.

The electrochemical impedance spectroscopy (EIS) of the assembled lithium-ion battery was also measured in more detail. As shown in Appendix A, the EIS curves of the cube Fe_2_O_3_ and the Fe_3_O_4_@void@N-doped C-*x* composite are composed of a small semicircle and a straight line. The small semicircle in the high frequency region is related to the charge transmission process of the electrons and lithium ions at the conductive junction. The lines in the low-frequency region are related to the solid diffusion process of the lithium ions in the active material. It can be observed that the diameter of the small semicircular cube Fe_2_O_3_ is larger than that of the Fe_3_O_4_@void@N-doped C-*x* composite, indicating that its electrical conductivity is worse than that of the Fe_3_O_4_@void@N-doped C-*x* composite. The good conductivity of the composites can be attributed to the carbon coating formed by dopamine, which contains many unsaturated bonds (such as C=N and C=C bonds) that are able to provide a large number of excellent conductive matrices for Fe_3_O_4_. In addition, the lines in the low-frequency region are all close to 45^o^, indicating that the lithium ions have good diffusion ability in the active materials.

## 3. Conclusions

In this work, PDA was used as a carbon source to coat Fe_2_O_3_ cubes to form a nitrogen-doped carbon coating layer. Further, the acid etching method was used to prepare composite materials with a core–shell structure and with different void sizes to act as the electrode materials in a lithium-ion battery. The electrochemical test results show that the specific capacity of the composites did not increase as the void size increased. Among all of the samples, the Fe_3_O_4_@void@N-doped C-5 with the appropriate void size showed the largest specific capacity, the best cycling performance, and the best rate performance. At a current density of 200 mA g^−1^, the discharge capacity of the Fe_3_O_4_@void@N-doped C-5 was able to reach 1222 mA h g^−1^ after 100 cycles, which is much higher than the levels achieved by the cubic Fe_2_O_3_. The appropriate size of the void between the carbon shell and the Fe_3_O_4_ core is not only beneficial to alleviate the volume expansion of Fe_3_O_4_ during the cycle, but also to electrolyte diffusion and the transmission of lithium ions.

## Data Availability

Not applicable.

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
