# Peer review of "Preparation of Hollow Core–Shell Fe_3_O_4_/Nitrogen-Doped Carbon Nanocomposites for Lithium-Ion Batteries"

_molecules, 2022, doi:10.3390/molecules27020396_

Round 1
Reviewer 1 Report
1. Authors claim that "Compared with other metal oxides, FeOx has the advantages of low price, abundant resources and environmental friendliness" - it would be better if authors talk with some specific oxides than iron oxide.
2. Is there any co relation between the battery like performance of iron oxide and magnetic property?
3. Magnetic properties of iron oxide nanoparticles must be performed.
4. The ionization of iron oxides may also be evaluated during Cyclic voltammetry study.
Reviewer 2 Report
Questions Regarding Manuscript Submission to “Molecules”
The work is relatively “NEW” for Li-ion battery research. The writing of this paper is very good. The manuscript contains some interesting achievements, and the results are supported with sufficient discussions. I would like to recommend it for publication in Molecules. Nevertheless, some issues need to be solved before its acceptance and publication, as listed below:
- Figure S1: Please add hkl values and remove Chinese letter from fig S1 (a)
- Figure 5c: Please present dQ/dV curves for C/D profiles during cycling.
- At what potential did you perform EIS measurements?
- It is good to see EIS test is carried out in this work. The lithium diffusion coefficient, which is an important kinetics parameter, could be calculated from the EIS test result. Please add the equivalent circuit for fitting the Nyquist plots. The range is too wide and the semi-circles are not visible.
- Language needs minor revision
Reviewer 3 Report
Recommendation: minor revision.
Comments: In this work, the authors reported core-shell Fe3O4@void@N-Doped C nanocubes as anode for lithium-ion batteries via hydrothermal method combined with the etching process. The morphology of these samples is beautiful and the electrochemical performance is good. The experiment data relevant to the interconnected Fe3O4@void@N-Doped C offered in this manuscript are sufficient to support the conclusion. Therefore, I recommend that this manuscript can be accepted after minor revision.
- The carbon content in these Fe3O4@void@N-doped C-x composite may be revealed by TGA analysis.
- The SEM characterization of Fe3O4@void@N-doped C after cycles may add to investigate the structuralstability and this anode undergoes during the charge/discharge process.
- TheCV analysis for the charge/discharge mechanism on page 5 maybe is cited some reference.
- The authors should compare the electrochemical performance with reported Fe3O4-based anode materials.Such as Eur. J., 2020, 26, 14708-14714; Adv. Energy Mater., 2016, 6, 1600256; ACS Appl. Mater. Interfaces, 2016, 8, 26878-26885. J Phys Chem C, 2019, 123(20): 12614-12622. Small, 2017, 13(33): 1701275.
Round 2
Reviewer 1 Report
The m/s may be accepted for publication.